# A Microfluidic Approach for Probing Heterogeneity in Cytotoxic T-Cells by Cell Pairing in Hydrogel Droplets

**DOI:** 10.3390/mi13111910

**Published:** 2022-11-04

**Authors:** Bart M. Tiemeijer, Lucie Descamps, Jesse Hulleman, Jelle J. F. Sleeboom, Jurjen Tel

**Affiliations:** 1Laboratory of Immunoengineering, Department Biomedical Engineering, Eindhoven University of Technology, 5600 MB Eindhoven, The Netherlands; 2Institute for Complex Molecular Systems, Eindhoven University of Technology, 5600 MB Eindhoven, The Netherlands; 3Microsystems, Department of Mechanical Engineering, Eindhoven University of Technology, 5600 MB Eindhoven, The Netherlands

**Keywords:** single-cell, CD8 T-cell, cytokines, droplet microfluidics, microgel, heterogeneity

## Abstract

Cytotoxic T-cells (CTLs) exhibit strong effector functions to leverage antigen-specific anti-tumoral and anti-viral immunity. When naïve CTLs are activated by antigen-presenting cells (APCs) they display various levels of functional heterogeneity. To investigate this, we developed a single-cell droplet microfluidics platform that allows for deciphering single CTL activation profiles by multi-parameter analysis. We identified and correlated functional heterogeneity based on secretion profiles of IFNγ, TNFα, IL-2, and CD69 and CD25 surface marker expression levels. Furthermore, we strengthened our approach by incorporating low-melting agarose to encapsulate pairs of single CTLs and artificial APCs in hydrogel droplets, thereby preserving spatial information over cell pairs. This approach provides a robust tool for high-throughput and single-cell analysis of CTLs compatible with flow cytometry for subsequent analysis and sorting. The ability to score CTL quality, combined with various potential downstream analyses, could pave the way for the selection of potent CTLs for cell-based therapeutic strategies.

## 1. Introduction

Cytotoxic T-cells (CTLs) are specialized cells that recognize and kill malignant or infected cells, making them the commander-in-chief and the soldiers of the immunological army. The efficiency of these CTLs to find and successfully kill their targets is dependent on the well-coordinated activation of naïve CTLs in the lymph nodes. Only upon encountering antigen-presenting cells (APCs) in the lymph node will they turn into effector CTLs, after which they will return to circulation and actively seek and destroy target cells at sites of inflammation in the tissue. Importantly, a few studies revealed that disease control depends on CTL quality rather than on quantity, where quality was defined by multiple functions [1,2,3]. In line with that, several sub-populations of CTLs were identified based on secreted cytokines and the presence of polarizing cytokines [4,5], suggesting that specific subsets of CTLs are relevant for immunity. Since interrogation of each individual cell is needed to truly identify CTLs of interest, the field has recently moved from conventional bulk experiments to single-cell approaches. Over the past two decades, single-cell technologies have emerged with the goal to dissect cellular heterogeneity and interrogate relevant sub-populations [6,7,8,9,10]. To efficiently and reproducibly probe the heterogeneity upon CTL activation, microfluidic tools were developed to allow precise manipulation and compartmentalization of single cells in small volumes (pico- to nano-liter) [11], thereby providing highly controlled environments acting like bioreactors to efficiently activate CTLs [12,13]. Although these efforts to pair CTLs either with target cells or APCs in microwell- and microtrap-based devices have proven to be effective, they are often restricted by a limited throughput since pairs are captured on-chip and the sample size is thus restricted by the dimensions of the device [12,14]. Droplet-based microfluidics has the major advantage that the sample size can be increased by longer running times thereby yielding higher throughputs [15,16,17,18], which is especially important when one is studying heterogeneity in immune cell responses and rare cell behavior is expected. Previous work aimed at pairing and studying CTL responses in droplets yielded interesting findings at a single-cell resolution [19,20]. However, these studies were often limited to the measurement of only a few parameters through microscopy imaging. As various factors, including both membrane marker expression and cytokine secretion, were previously used to score CTL quality, combining these will allow for true interrogation of the heterogenous CTL response [1,2,3]. To overcome both the challenge of limited throughput as well as allowing measurement in a multiparameter fashion, hydrogels can be incorporated during droplet production to obtain “microgels”, in which cells can be paired. These will maintain the spatial coupling of single-cell pairs while allowing subsequent cell profiling using flow cytometry, facilitating a multiparameter measurement [21]. Hence, microgels could pave the way for novel and robust analytic tools to study single CTL and APC interactions [12,22,23].

Here, we generated a single-cell droplet microfluidics platform to probe CTL heterogeneity upon activation using soluble stimuli and artificial antigen-presenting cells (aAPCs). We incorporated ultra-low melting point agarose hydrogels to create microgels containing CTL/aAPC pairs for downstream analysis by flow cytometry. Activation of CTLs was measured in a multidimensional fashion, screening both expression of membrane markers as well as multiplexed cytokine secretion. We observed distinct CTL activation profiles induced by soluble stimuli compared to aAPCs. This approach allows for scoring the quality of CTL activation combined with various potential downstream applications after sorting, and can therefore greatly benefit future immune cell therapeutic applications.

## 2. Materials and Methods

### 2.1. Cell Isolation and Preparation

CD8+ T-cells were isolated from buffy coats obtained from healthy human donors (Sanquin bloodbank, Eindhoven, The Netherlands) after written informed consent per the Declaration of Helsinki and according to the institutional guidelines. Peripheral blood mononuclear cells were isolated using Lymphoprep (Stemcell Technologies, Vancouver, Canada) according to manufacturer’s protocol, after which CD8+ cells were isolated using magnetic-activated cell sorting kit (Miltenyi Biotech, Bergisch Gladbach, Germany). Isolated CD8+ T-cells were resuspended in RPMI medium (Gibco, Life Technologies, Carlsbad, CA, USA) with 2% Human Serum (Sanquin Bloodbank) and 1% Penicillin–Streptomycin (Gibco, Life Technologies), hereafter referred to as culture medium. The cells were then coated with capture antibodies for IL-2, TNF-α, and IFN-γ (Miltenyi Biotech). After incubation and washing, the cells were resuspended in culture media for droplet encapsulation.

### 2.2. Microfluidic Device Fabrication

Droplet microfluidics devices were produced using soft lithography. Photomasks were ordered from CAD/Art Services, Inc. (Bandon, OR, USA). PDMS molds were produced by spin-coating wafers with SU-8 3000 photoresist (Microresist Technology, Berlin, Germany) according to manufacturer’s protocol to obtain 30 µm of channel height. PDMS devices were fabricated by mixing SYLGARD^®^ 184 PDMS with SYLGARD^®^ 184 curing agent (both from Merck) at 10:1 *w*/*w* before pouring the mixture onto the PDMS molds and curing for 2 h at 65 °C. Using a 1 mm biopsy puncher, holes for the inlets and outlet were punched. The obtained PDMS devices were bonded to glass slides using a plasma asher (Emitech, K1050X, Montigny-le-Bretonneux, France). After bonding, the channels were treated with 5% perfluorooctyltriethoxysilane in HFE-7500 fluorinated oil (both from Fluorochem, Hadfield, United Kingdom) in order to make channel walls hydrofobic, incubated for 1 h at 65 °C, flushed again with HFE-7500, and incubated overnight at 65 °C for thermal bonding.

### 2.3. Production of Temperature Regulation Device

The designs for all heating devices were made in Siemens NX (Siemens AG, Munich, Germany) (designs are available as Electronic Appendix A). All devices were printed in clear resin (RS-F2-GPCL-04, Formlabs, Somerville, MA, USA) using a Formlabs Form 3 SLA printer. After printing, the uncured resin was removed from the channels by flushing them several times with clean isopropanol from a 20 mL syringe that was directly connected to the Luer-lock connections. The prints were then washed and cured per the manufacturer’s instructions in a Form Wash (FH-WA-01, Formlabs) and Form Cure (FH-CU-01, Formlabs) station. The chip platform was finished by bonding a 0.5 mm PMMA plate to the bottom using super glue (Loctite, Düsseldorf, Germany) and a glass microscope slide to the top using Dowsil™732 silicon glue (Dow Corning, Midland, MI, USA). Luer-lock to barb connectors (Cole-Parmer, Vernon Hills, IL, USA) were used to connect the tubing. The three heating devices were connected to a water pump (7026898, RS PRO) in series and warm water was flushed through to maintain a temperature of 37 °C.

### 2.4. Droplet Production, Cell Encapsulation and CTL Stimulation

Droplet production was performed using a previously reported pipette tip method [24], and by attaching the droplet device to a neMESYS microfluidic pump (Cetoni, Korbußen, Germany). The first inlet was used for 2.5% Picosurf (Spherefluidics, Cambridge, United Kingdom) in HFE-7500 (30 µL/min), the second for CD8+ T-cells (5 µL/min), and the last for stimuli, or aAPC Dynabeads (Thermofisher) (5 µL/min). Cells were injected at a concentration of 4 × 10^6^ cells/mL. When Dynabeads were included these were injected at a concentration of 1 × 10^7^ particles/mL. Soluble stimuli were added at a concentration of 1 mg/mL for PMA and 10 mg/mL for ionomycin (both from Peprotech). When producing aqueous droplets cells, Dynabeads and stimuli were suspended in culture media, when producing microgels they were suspended in culture media containing 1% *w/v* of ultra-low melting point agarose (Merck, Kenilworth, NJ, USA). Droplets were collected in Eppendorf tubes and incubated at 37 °C and 5% CO_2_ for 24 h.

### 2.5. Droplet Characterization

To determine the distribution of cells among droplets, contents were manually counted in brightfield microscopy-obtained images. Droplet size was determined using ImageJ software [25], where automated thresholding was used to create a greyscale image of the brightfield images after which the particle analysis function was used to automatically measure droplets. The diameter was calculated as the average between the major and minor axes of detected particles.

### 2.6. Cell/Microgel Retrieval and Flow Cytometric Measurement

After 24 h of incubation, droplets containing agarose were cooled to 4 °C for 30 min to obtain microgels. Both cells and microgels were retrieved from the emulsion by adding 20% of 1H,1H,2H,2H-perfluoro-1-octanol (PFO) in HFE-7500 onto the emulsion at a 1:1 *v*/*v* ratio. The obtained solution was washed and afterward stained using Zombie NIR viability kit (Biolegend). Next, they were stained with a cocktail of cytokine-detection antibodies for IFNγ, TNFα, and IL-2 (all from Miltenyi Biotech), along with antibodies to detect surface marker expression; CD8-Brilliant violet 605, CD69-Brilliant violet 650, and CD25-Brilliant violet 786 (all from Biolegend), according to manufacturer’s protocols. During the staining of microgels, the incubation time was doubled to give antibodies more time to diffuse into the agarose. When washing microgels, the washing solution was kept on the microgels for 5 min before spinning down. Centrifugation of microgels was performed at 100 RCF for 10 min. After the staining procedures, fluorescent values of both cells and microgels were measured using FACSymphony (BD) and data were analyzed using Flowjo Software 10.7.0 (FLowJo LLC, Ashland, OR, USA).

### 2.7. Statistical Analysis

Data processing and statistical analysis were performed using PRISM 9 (Graphpad software). Data are shown as mean ± standard error of the mean (SEM) unless indicated differently. Statistical analysis was performed using repeated-measures one-way ANOVA with post-hoc Tukey’s test after normality was proven using Shapiro–Wilk test. *p* < 0.05 was considered significant.

## 3. Results

### 3.1. Single CTL Activation in Droplets Reveals Highly Heterogeneous Responses

Stimulation of single cells with soluble stimuli has proven a great way to reveal heterogeneous behavior using droplet-based microfluidic platforms [26,27,28]. Droplets provide controlled environments containing cells along with stimuli and assay reagents, thereby preventing cells from influencing each other in a juxta- or paracrine fashion. We investigated primary human CTLs and encapsulated them in 70 picolitre-sized droplets along with Phorbol 12-myristate 13-acetate (PMA) and ionomycin as stimuli [13,29] (Figure 1A). Microscopy analysis showed that the cell encapsulation followed the predicted Poisson distribution (Appendix A) [30] and ensured that virtually all CTLs were encapsulated as single cells in monodisperse droplets. After single-cell culture, CTLs were retrieved from droplets by PFO-induced de-emulsification. Subsequently, CTL activation was assessed by the early activation markers CD69 and CD25. CD69 is an early inducible cell surface glycoprotein acquired during activation and functions as a signal-transmitting receptor [31]. CD25 is the alpha chain of the trimeric IL-2 receptor and is considered to be a prominent early-to-middle cellular activation marker [32]. Simultaneously, we investigated the secretion of the cytokines TNFα, IFNγ, and IL-2, captured via membrane-bound constructs. We observed three distinct phenotypes based on the expression of activation markers CD25 and CD69 (Figure 1C,D). Within these populations, a high degree of heterogeneity was found with respect to the secreted cytokines (Figure 1E). These encompassed most combinations possible, confirming the heterogeneous nature of the CTL activation. Interestingly, membrane marker upregulation appeared reversely correlated with cytokine secretion as the CD69-CD25- population appeared to have the lowest percentage of non-producing cells. Furthermore, the CD69-CD25- population showed an increased tendency to produce a combination of IFNγ and TNFα when compared to the other two populations. These results indicate the analytical potency of our droplet platform. CTL activation via PMA/Ionomycin is often used in the literature; however, it surpasses the natural way of CTL activation by TCR engagement and as such does not mimic a physiological setting. Thus, an activation model more resembling immune activation via direct cell contact would translate much better to any system of therapeutic value.

### 3.2. Spatial Pairing Data Are Lost in Aqueous Droplets upon CTL and aAPC Interactions

In order to achieve effective CTL activation, APCs need to engage with naïve CTLs in the lymph nodes. Activation involves three main signals [33,34]. The first being the TCR/CD3 complex binding to the peptide antigen presented by the major histocompatibility complex of APCs. Co-stimulatory molecules provide the second signal [35,36] by binding to the CD28 receptors located in close proximity to the TCR. Together, these signals lead to the release of cytokines, which further shape the development of the immune response [37]. To mimic the first two signals of this cell–cell contact-dependent activation in our platform, we aimed to co-encapsulate CTLs with aAPCS (Figure 2A,B and Appendix A), which contain anti-CD3 and anti-CD28 antibodies and are routinely used to activate T-cells both in bulk as well as at single-cell level [38,39]. By switching from soluble stimuli to aAPCs, the encapsulation efficiency is altered but the prevalence of all combinations of encapsulation still followed the predicted Poisson distribution (Figure 2B and Appendix A). After in-droplet co-culture of single-cell pairs and verification of CTL/aAPC interaction based on cell morphology (Supplementary Video S2), we assessed CTL activation and observed heterogeneous expression of CD69 and CD25 (Figure 2C), and very little cytokine secretion in only a small percentage of CTLs (Figure 2D). Unlike in the PMA/ionomycin stimulation, here, we cannot ensure that individual cells are co-encapsulated with aAPCs. According to the Poisson distribution, the fraction of encapsulated CTLs with one or multiple aAPCs can be estimated at around 50% but only around 30% showed an increase in membrane marker expression. Therefore, the preservation of spatial coupling of a CTL and aAPC would be an elegant approach to maintain such information until analysis of activation.

### 3.3. Agarose Microgels as Bioreactors for Subsequent Flow Cytometry Measurement

To achieve spatial coupling of CTLs with aAPCs, hydrogel-based microgels were desired with properties that remain soluble during droplet production and culture but have a trigger-cross-linked ability. Ultra-low melting-point agarose proved an ideal candidate as it remains soluble under culture conditions at 37 °C and crosslinks below 18 °C prior to downstream analysis [40]. Advantages are pore size >~200 nm warranting diffusion of detection antibodies, and biocompatibility [27,41]. By using 3D-printed devices (Figure 3B), we ensured the encapsulation of CTLs in agarose solutions under temperature-controlled conditions to avoid clogging of the device by premature gelation, resulting in monodisperse droplet formation (Figure 3C). After cross-linking and de-emulsification, monodisperse agarose microgels containing cells were retrieved in PBS free of oil (Figure 3C). By comparing flow cytometric measurement of cells in microgels and empty microgels with respect to unencapsulated cells, the microgels appeared to primarily increase the scattering (SSC-A) of measured events (Appendix A). Microgels containing CTLs were selected by intra-microgel staining for CD8 membrane protein (Figure 3D). We investigated whether the pore size of agarose hydrogels allowed diffusion of fluorophore-conjugated antibodies to ensure that CTL activation in microgels could be detected by our antibody panel. CTLs were therefore activated in bulk using PMA and ionomycin, partially stained in microgels, and partially stained in bulk. Altered expression, compared to unstimulated controls, indicated that intra-microgel and cell-specific staining of CTLs was achieved (Figure 3E). Additionally, microscopic images showed that staining was cell-specific and did not differ between encapsulated and unencapsulated cells (Appendix A). Even for IL-2 detection, which uses the largest fluorophore–antibody conjugate [42], only a small difference could be observed in fluorescent intensity. Thus, microgel-encapsulated cells are compatible with fluorescent staining for flow cytometric analysis. Future additions of fluorophores can be reliably incorporated to extend the antibody panel for multidimensional analysis.

### 3.4. Phenotypic and Functional Analysis of CTL/aAPC Pairs in Microgels

Having established that detection of activation markers in microgels is possible, we moved forward to induce cell contact-mediated CTL activation. First, we monitored the location of cells and aAPCs to ensure that cell–bead interactions are not affected by agarose viscosity and still take place such as in aqueous droplets [43,44]. Brightfield images were captured during experiments directly after droplet production and again after 24 h of culture followed by agarose crosslinking (Figure 4A). Analysis of 532 droplets containing CTL/aAPC pairs demonstrated that single CTLs effectively interacted with aAPCs (Figure 4B), excluding that agarose viscosity affected cellular interactions. Additional temporal monitoring over the first 30 min after encapsulation was performed as well and displayed cells latching onto aAPCs within minutes, demonstrating that cell–cell interaction takes place, even in agarose droplets (Appendix A). Next, we selected all microgels which contained CTLs based on CD8 expression (Figure 4C) and subsequently checked pairing with aAPCs by autofluorescent properties of aAPCs. The position of pairs containing a single aAPC was verified based on overlaying the autofluorescence of a single aAPC (Appendix A). The gated populations were compared and indeed indicated that 1:1 ratio single-cell contact-mediated activation occurred in a percentage of cells according to membrane markers (Figure 4D) as well as cytokines (Figure 4E). Furthermore, the observed percentages of activation approximated the percentages observed under aqueous conditions (Figure 2C,D). Especially when taking into account that in those experiments, ~50% of all measured cells actually encountered an aAPC. This demonstrated that agarose encapsulation does not affect mechanisms of activation and successfully serves its purpose of maintaining spatial information. Moreover, we could clearly distinguish microgels where CTLs were paired with one aAPC or multiple aAPCs and in this way probed the effect of multiple interactions. We demonstrated that multiple interactions resulted in an increased number of activated CTLs, primarily based on CD69 and CD25 marker expression (Figure 4E). Besides pairing of CTLs with beads, the platform can easily be adjusted to pair CTLs with another cell type. This would merely require the incorporation of an additional membrane marker staining, cell pairs can then be selected by gating for double-positive events (Appendix A). Taken together, these results demonstrate that agarose encapsulation is a highly potent approach to investigating single-cell contact-mediated activation of CTLs in a high-throughput and multiparameter fashion.

### 3.5. Single-Cell Decoding of CTL Activation and Secretion Based on Different Stimuli

We investigated CTL activation using PMA/Ionomycin and aAPC-interaction as stimulation models at the single-cell level. This allowed us to establish a novel depth of comparing single-cell heterogeneity, based on both cell phenotype and function. Both models showed similar patterns in CD69 and CD25 expression, with differences in frequencies (Figure 1E and Figure 4D). The most prevalent secreted cytokines within these populations were; (1) IFNγ in combination with TNFα; (2) IFNγ, TNFα and IL-2 in PMA/Ionomycin-stimulated CTLs (Figure 5A); (3) IFNγ in combination with TNFα and (4) TNFα only in aAPC-stimulated CTLs (Figure 5B). However, the correlation between membrane markers and cytokine production was vastly different between the two activation models. In PMA/Ionomycin-stimulated CTLs, membrane marker expression and cytokine production appeared to be negatively correlated since the double-negative phenotype (CD69-CD25-) showed the least amount of non-producing cells. On the contrary, membrane marker expression and cytokine production appear to be positively correlated in aAPC-stimulated CTLs, with the least amount of non-producing cells observed in the double positive phenotype (CD69+CD25+). Additionally, IL-2 secretion appears much more prevalent in the PMA/Ionomycin stimulated conditions, primarily when all three cytokines are secreted. These findings underline the difference between the two models, where PMA/ionomycin is a non-specific synthetic approach to obtain optimal cytokine secretion, and aAPCs are used to mimic biologically relevant mechanisms. Nevertheless, to the best of our knowledge, this difference has not been previously observed at this degree of single-cell resolution, which underlines the strengths of the demonstrated platform.

## 4. Discussion

We developed a droplet microfluidic-based single-cell activation platform for the screening of cytotoxic T-cell activation. In its standard form, the platform allowed us to activate CTLs at the single-cell level with synthetic soluble stimuli and detect heterogeneity based on multiparameter measurement of phenotype and function. In its more advanced form, we co-encapsulated CTLs and aAPCs in agarose microgels to study single-cell contact-mediated activation between single cells. In particular, microgel encapsulation allowed us to highlight the difference in CTL activation mechanisms upon stimulation with PMA/Ionomycin and aAPCs, whilst obtaining novel resolution over the resulting heterogeneous responses. Therefore, this platform can be used to dissect CTL multifunctionality based on the profiling of secreted cytokines, which is an important facet of strong adaptive immunity.

In comparison to previous research on droplets for cell pairing [20,44,45,46,47,48], our approach warrants unlimited throughput and is not limited by droplet capture in traps, wells, or observation chambers. Additionally, such methods often rely on monitoring via microscopy. Although this allows for temporal resolution, it limits the number of read-out parameters [49,50]. In applications where multiparameter functional measurements are possible, the ability to recover interrogated cells for downstream applications is often limited [51]. In our approach, by sacrificing temporal resolution, we are able to measure cell pairs using flow cytometry, which opens up the possibility for multiparameter (potentially up to 45 markers) measurement as well as sorting for downstream applications, which was previously reported with agarose microgels [52,53,54,55,56]. Combining single-cell analysis with fluorescence-activated droplet sorting was previously performed on-chip, where droplets are sorted either before [48,57] or after culture [19,47]. These works have great potential but are also limited by the number of measured parameters and the closed system they are performed in. For example, Gerard et al. demonstrated very potent sorting of specific IgG-secreting cells, but their sort is based on only a single parameter and adapting it for a different application will change the entire system [47]. Conversely, recovering cell pairs in microgels offers more flexibility towards fluorescent staining after culture, as well as the use of well-developed commercially available flow cytometers. Yanakieva et al. demonstrated this principle by sorting pairs of secretor cells and reporter cells in microgels in order to enrich yeast clones secreting biorelevant proteins [52]. However, such applications have previously only been demonstrated with yeast, bacteria or cell lines. Here, our microgel-based platform aims at monitoring the effect of physical cell–cell contact in primary immune cell activation. We achieved a higher number of screened parameters allowing for the study of both immune cell phenotype and functionality.

Nevertheless, a potential hurdle after the recovery of microgels is retrieving cells from encapsulation. For agarose microgels, this would require heating to above 70 degrees [40], or enzymatic digestion [58]. Potential candidates to avoid such harmful processes could be alginate [59,60,61] or thermo-reversible hydrogels [26,62]. In future adaptations, these could be utilized to extend our platform to cell retrieval. Furthermore, we demonstrated the compatibility of agarose microgels with downstream phenotypic and functional studies, but this approach could be even extended to genetics since agarose microgels have readily been shown to be suitable for PCR and sequencing purposes [63,64,65]. For example, single-cell sequencing could be performed to enable the study of TCR sequences which were shown to have high relevance and potential to design better vaccines or autoimmune therapies [66,67]. Reversely, the platform can be utilized to study the heterogeneity of APCs [10] as the panel of markers that is measured using flow cytometry can be readily switched to include different targets of interest. The potential of our platform is highlighted by the interesting observed differences in CTL activation approaches. This interesting difference might be explained by the different mechanisms exploited by the two activation models. On one hand, aAPCs target CD3 and CD28 receptors on the CTLs, partially mimicking how naïve cells are activated by APCs in the lymph node. On the other hand, PMA directly targets protein kinase C (PKC) and Ionomycin upregulates intracellular calcium, thus synergizing with PMA to activate PKC and completely bypassing membrane receptors [29,68]. PMA/Ionomycin is therefore much less physiologically relevant and might result in less intuitive and relevant results from a T-cell biology perspective.

Taken together, we believe that all these options illustrate the flexibility and potential of this droplet-based platform to investigate the activation of the adaptive immune system at the single-cell level.

## Figures and Tables

**Figure 1 micromachines-13-01910-f001:**
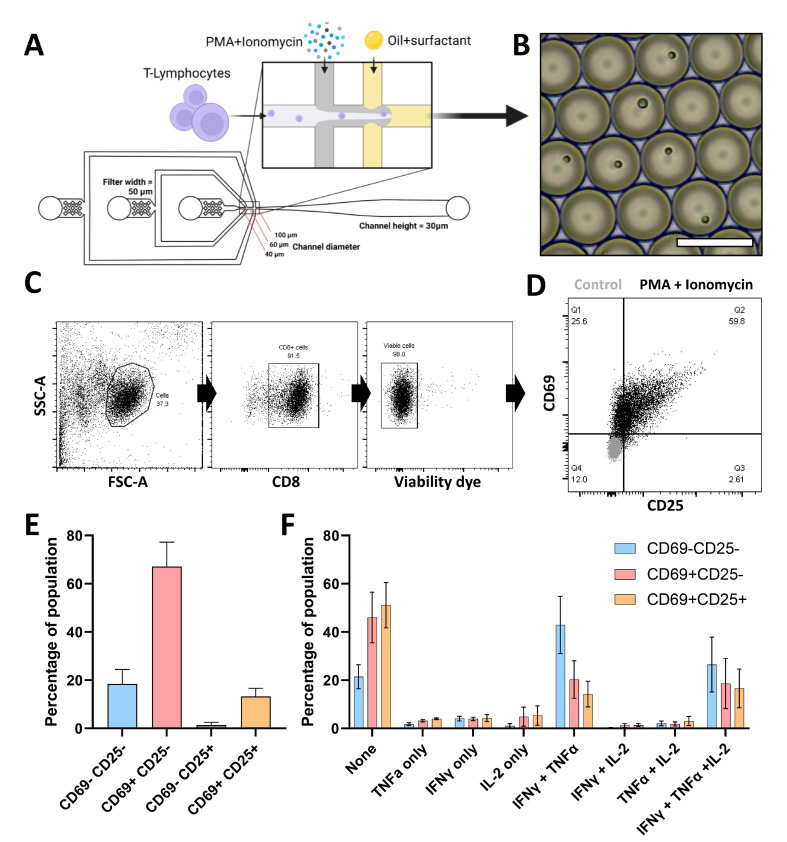
Single-cell CTL activation using soluble stimuli. (**A**) Layout of the microfluidic device used for droplet production including popped-out schematic of droplet formation, scale bar represents 500 µm. Layout is a direct copy of the photolithography mask design, and channel height is equal throughout. (**B**) Brightfield microscopy image of aqueous droplets containing single CTLs. Scale bar = 100 µm. (**C**) Gating strategy for flow cytometry data to select viable CD8-positive cells. (**D**) Marker expression of selected viable CD8-positive cells for CD69 and CD25 activation markers, either after stimulation with PMA + Ionomycin (black dots) or unstimulated (grey dots). Data display one representative donor. (**E**) Prevalence of populations of PMA + Ionomycin activated CTLs based on CD25 and CD69 expression. Data represent SEM of *n* = 4 biological replicates. (**F**) Bar graph displaying the frequency of different profiles of secretion as exhibited by the three prevalent populations of PMA + Ionomycin stimulated CTLs. Data represent average values of *n* = 3 biological replicates.

**Figure 2 micromachines-13-01910-f002:**
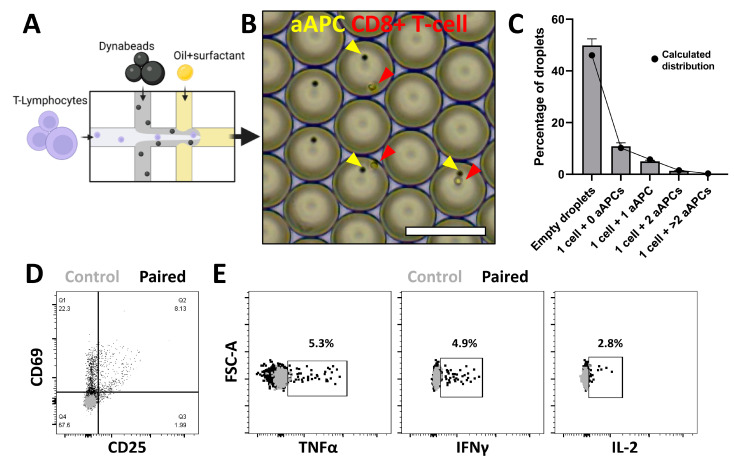
Single-cell CTL activation using aAPCs. (**A**) Schematic of droplet formation when CTLs are paired with Dynabeads. (**B**) Brightfield microscopy image of aqueous droplets containing CTLs (red) and aAPCs (yellow) in pairs. Scale bar = 100 µm. (**C**) Counted (grey bars) and calculated (black dots) Poisson distribution of droplet contents when co-encapsulating CTL and aAPCs. Error bars represent average ± SEM of 4 independent experiments in which at least 1000 droplets were counted. (**D**) Marker expression of CD69 and CD25 activation markers on CTLs retrieved from droplet co-culture with aAPC. (**E**) Cytokine secretion of CTLs retrieved from droplet co-culture with aAPCs (black dots) as compared to unstimulated control (grey dots).

**Figure 3 micromachines-13-01910-f003:**
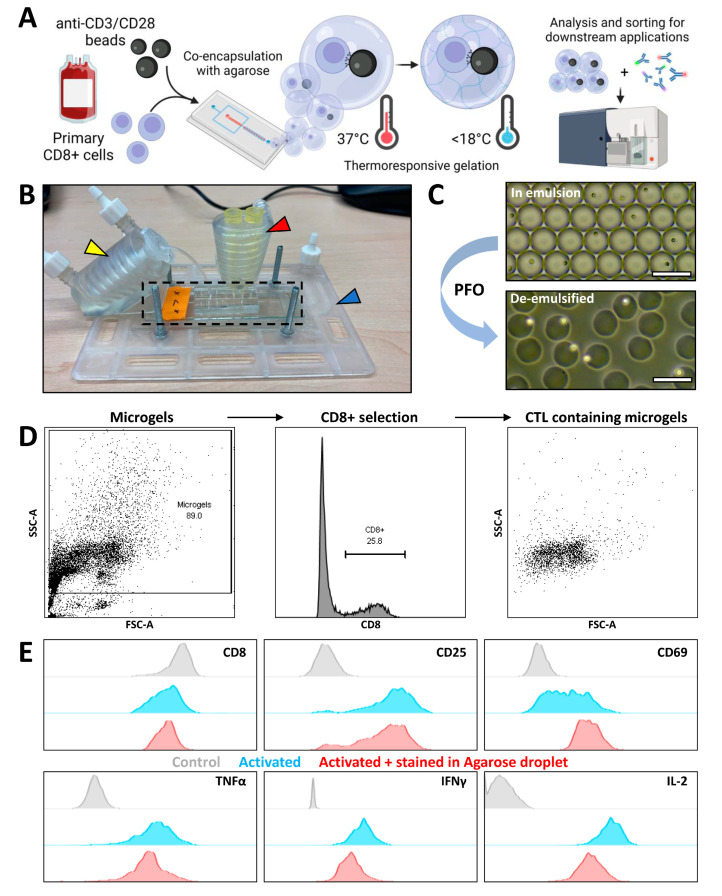
Encapsulation and flow cytometric measurement of CTL-containing agarose microgels. (**A**) Schematic of the workflow; cells are isolated and co-encapsulated with aAPCs in droplets along with agarose solution, droplets containing cells are cultured at 37 °C to allow interaction after which temperature is decreased below 18 °C to allow agarose to crosslink and create microgels containing cell pairs, microgels are retrieved and can be stained for membrane markers and measured or sorted using flow cytometry. (**B**) 3D printed temperature control devices. Device used to control temperature in pipette tips (red arrow) for injection of samples, microscope insert (blue arrow) containing microfluidic device (black dashed line), and heated holder for collection tube (yellow arrow). Hot water is flown through all devices in series to maintain 37 °C during droplet formation and after collection. (**C**) Brightfield microscopy images of water-in-oil emulsion before crosslinking and of de-emulsified microgels in PBS. Scale bars = 100 µm. (**D**) Gating strategy to retrieve cell-containing droplets; Microgels are gated based on the FSC/SSC of empty droplets, CD8+ events are selected. (**E**) Histograms depicting the effect of antibody staining in agarose hydrogels compared to staining in conventional cell solution. Comparing unstimulated control stained in solution (grey), PMA/Ionomycin bulk stimulated cells stained in solution (blue), PMA/Ionomycin bulk stimulated cells stained in microgels (red).

**Figure 4 micromachines-13-01910-f004:**
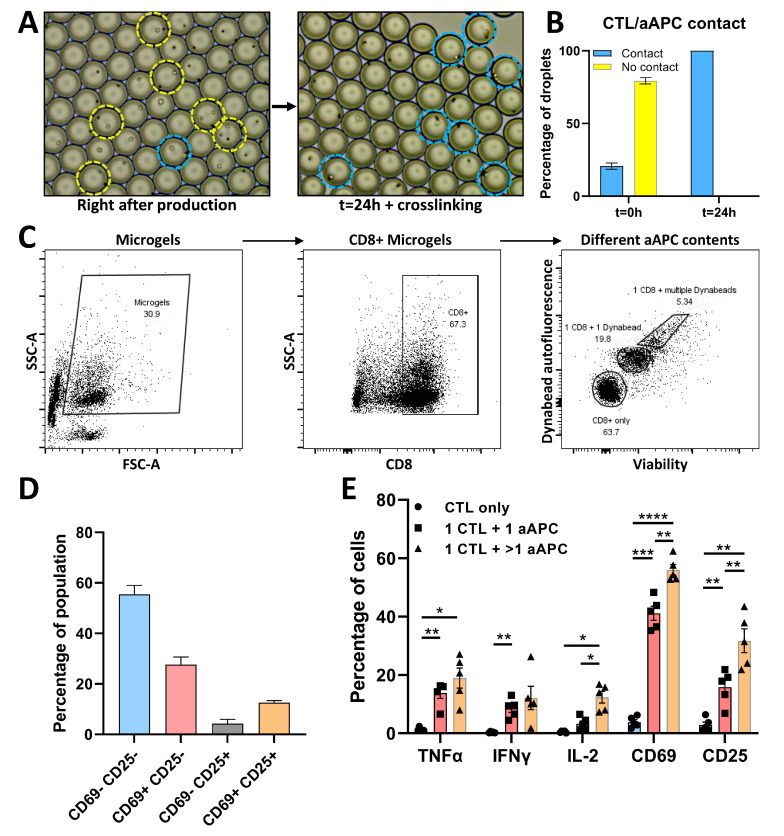
Encapsulation and flow cytometric measurement of CTL/aAPC pairs in microgels. (**A**) Brightfield microscopy images of droplets for CTL/aAPC pairing; Left image displays droplets directly after production at 37 °C where cells and particles are free floating in agarose solution droplets, right image displays droplets after 24 h of incubation at 37 °C and subsequent gelation at >18 °C where cells are fixed in crosslinked agarose. Yellow circles indicate CTL/aAPC combinations where no contact is observed, blue circles indicate combinations where contact is observed. (**B**) Results of manual quantification of *n* = 46 images containing *n* = 532 CTL/aAPC paired droplets from *n* = 3 independent experiments. error bars represent *n* = 3 experiments. (**C**) Gating strategy of CTL/aAPC droplets; all non-empty microgels are selected, all microgels containing at least a CD8+ cell are gated, based on Dynabead autofluorescence and viability dye intensity gates are drawn for single viable CTLs, 1:1 pairs of CTL/aAPCs, and microgels containing a CTL and multiple aAPCs. (**D**) Prevalence of populations of 1:1 aAPC activated CTLs based on CD25 and CD69 expression. Data represent SEM of *n* = 5 biological replicates. (**E**) Data summary of activation markers in microgels containing only CTLs (blue), microgels containing 1 CTL and 1 aAPC (red), and microgels containing 1 CTL with multiple aAPCs (orange). Data show *n* = 5 biological replicates with SEM. * *p* < 0.05, ** *p* < 0.01, *** *p* < 0.001, **** *p* < 0.0001, if no significance is indicated none was found.

**Figure 5 micromachines-13-01910-f005:**
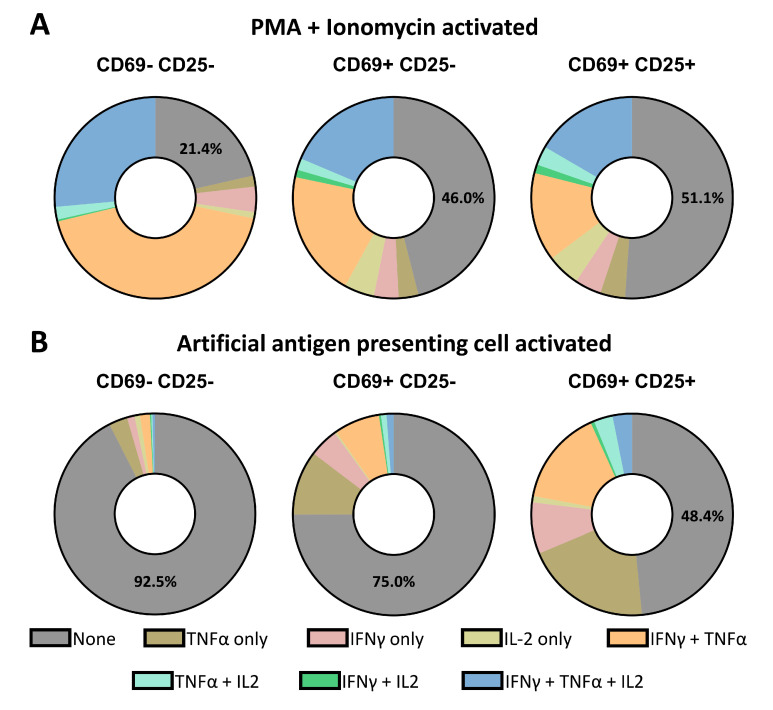
Comparing heterogeneous cell responses between PMA/Ionomycin and aAPC activation. (**A**) Pie graphs displaying the different profiles of secretion after PMA/Ionomycin stimulation of CTLs, *n* = 3 donors. (**B**) Pie graphs displaying the different profiles of secretion after aAPC stimulation of CTLs, *n* = 5 donors.

## Data Availability

The raw data supporting the conclusions of this article will be made available by the authors upon request.

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
