# Peer review of "A Microfluidic Approach for Probing Heterogeneity in Cytotoxic T-Cells by Cell Pairing in Hydrogel Droplets"

_micromachines, 2022, doi:10.3390/mi13111910_

Round 1
Reviewer 1 Report
In this manuscript, the authors demonstrated a single-cell droplet microfluidics platform for probing heterogeneity in cytotoxic T-cells (CTLs) that allowed deciphering single CTL activation profiles by multi-parameter analysis. The authors successfully built the microfluidic platform, and also realized the encapsulation of single cells and the functional heterogeneity probing of cytotoxic cells. However, there have been many reports on single cell analysis using microfluidic platforms, and the reviewer considers that this manuscript needs to provide more sophisticated data and detailed statements about the problems to be solved in the field to reflect the innovation of this work. Therefore, the reviewers recommended that the manuscript be major revised for publication in Micromachines. The problems to be solved are as follows:
1. There are many reports on single-cell analysis by microfluidic platform compatible with flow cytometry for subsequent analysis and sorting. Such as Nat. Biotechnol. 2020, 38, 715; Anal. Chem. 2019, 91, 12283; Angew. Chem. Int. Ed. 2018, 57, 236; Sci. Rep. 2017, 7, 1366. What are the innovations and superiority of this manuscript compared with other works?
2. In the introduction, the author did not state the problems existing in the study of functional heterogeneity of single-cell analysis of CTLs on microfluidic platform. The reviewer thought that just saying “But these studies were often limited by measurement of only few parameters per event or by absence of spatial pairing information” didn’t seem to make the point. It is suggested that the introduction section needs to be rethought to point out the existing problems in this field so as to highlight the theme and innovation points.
3. The cell encapsulation proportion in the microdroplet looks low in Figure 1b and Figure 2a. How to improve the encapsulation proportion of single cells, just by increasing the cell density does not seem to guarantee that each droplet contains a single cell? How to separate the microdroplets that are encapsulated with cells from unencapsulated cell droplets during testing? The number of microdroplets enveloping cells shown in Figure 2a is quite different from the statistical value in Figure 2b. Please provide more accurate optical photos.
4. An optical photo of staining should be shown in Figure 3 to prove that the pore size of agarose hydrogels allowed diffusion of fluorophore-conjugated antibodies to ensure that CTL activation in microgels.
5. In page 2, line 88-89, the authors described that “After bonding, the channels were treated with 5% perfluorooctyltriethoxysilane in HFE-7500 (both from Fluorochem)”. What does the HFE-7500 mean? Why does the channel need to be treated with 5% perfluorooctyltriethoxysilane in HFE-7500?
6. In page 3, line 112, the authors demonstrated that “Cells were injected at a concentration of 4*106 cells/ml”. “…4*106 cells/ml…” should be “… 4*106 cells/ml …”. Please pay attention to check the details and the specification of writing.
Author Response
Reviewer #1:
General comment: In this manuscript, the authors demonstrated a single-cell droplet microfluidics platform for probing heterogeneity in cytotoxic T-cells (CTLs) that allowed deciphering single CTL activation profiles by multi-parameter analysis. The authors successfully built the microfluidic platform, and also realized the encapsulation of single cells and the functional heterogeneity probing of cytotoxic cells. However, there have been many reports on single cell analysis using microfluidic platforms, and the reviewer considers that this manuscript needs to provide more sophisticated data and detailed statements about the problems to be solved in the field to reflect the innovation of this work. Therefore, the reviewers recommended that the manuscript be major revised for publication in Micromachines.
Response to general comments: We would like to thank the reviewer for their interest in our work, and for giving helpful feedback. The reviewer raised several aspects to add or revise the current manuscript, below we have addressed each of them:
Comment 1: There are many reports on single-cell analysis by microfluidic platform compatible with flow cytometry for subsequent analysis and sorting. Such as Nat. Biotechnol. 2020, 38, 715; Anal. Chem. 2019, 91, 12283; Angew. Chem. Int. Ed. 2018, 57, 236; Sci. Rep. 2017, 7, 1366. What are the innovations and superiority of this manuscript compared with other works?
Response to comment 1: We understand the reviewers concern that appropriate citing of previous work is important to allow readers to appreciate the novelty and innovations of published work. The papers suggested by the reviewer (Anal. Chem. 2019, 91, 12283; Angew. Chem. Int. Ed. 2018, 57, 236; Sci. Rep. 2017, 7, 1366) report the application of microfluidic platforms for cell co-culture or for cell counting. Importantly, these applications utilize either cell-lines or bacteria. Although these reports are very relevant in the development of single-cell technology, the applications are different from our presented work here, which addresses the characterization of primary immune cell interaction, phenotype and functionality on the single-cell level with the possibility of sorting hydrogel cell-pairs off-chip. However, the Nat. Biotechnol. 2020, 38, 715 article that the reviewer pointed out, indeed reports a powerful platform for single-cell phenotypic and genotypic analyses. This work was already included in our manuscript when enumerating previous works on droplets for cell pairing but perhaps not detailed enough. We therefore expanded the comparison between our work and other works in the discussion based on their compatibility with flow cytometry and sorting.
Changes in the revised version:
P12: “Combining single-cell analysis with fluorescence activated droplet sorting has been previously performed on-chip, where droplets are sorted either before48,57 or after culture19,47. These works have great potential but are also limited by the number of measured parameters and the closed system they are performed in. For example, Gerard et al. demonstrated very potent sorting of specific IgG-secreting cells, but their sort is based on only a single parameter and adapting it for a different application will change the entire system47. Conversely, recovering cell pairs in microgels offers more flexibility towards fluorescent staining after culture, as well as the use of well-developed commercially available flow cytometers. Yanakieva et al. demonstrated this principle by sorting pairs of secretor cells and reporter cells in microgels in order to enrich yeast clones secreting biorelevant proteins52. However, such applications have previously only been demonstrated with yeast, bacteria or cell-lines. Here, our microgel-based platform aims at monitoring the effect of physical cell-cell contact in primary immune cell activation. We achieved a higher number of screened parameters allowing for the study of both immune cell phenotype and functionality.”
Comment 2: In the introduction, the author did not state the problems existing in the study of functional heterogeneity of single-cell analysis of CTLs on microfluidic platform. The reviewer thought that just saying “But these studies were often limited by measurement of only few parameters per event or by absence of spatial pairing information” didn’t seem to make the point. It is suggested that the introduction section needs to be rethought to point out the existing problems in this field so as to highlight the theme and innovation points.
Response to comment 2: We would like to thank the reviewer for pointing out that this relevance was not yet sufficiently substantiated. T cells are generally already very heterogenous with specific T cell receptor expression, receptor affinity and avidity, and after activation a wide variety of effector functions are observed (10.1016/s0952-7915(03)00037-2). This is primarily the case after TCR-mediated activation via APC interaction. Variation can eventually be observed in terms of proliferation capacity and cytotoxic activity, or in earlier stages in regulatory functions such as secretion of cytokines (e.g. IL-2, IFNγ, TNFα) and expression of membrane markers (e.g. CD69 and CD25). As disease control has been shown to rely mostly on CTL quality rather than quantity, it is crucial to understand such heterogeneous behavior in response to TCR-mediated activation. Although, several studies have previously described pairing of primary human cells in droplets, very few studies have attempted CTL and APC pairing using droplets at the single-cell level (10.4172/2155-9899.1000334, 10.1038/s41419-020-03173-7). Furthermore, these studies were limited by monitoring of cell interactions on-chip by microscopy. This has major limitations such as measurements of only one or two parameters per event and restricted throughput per experiment. This last point is especially important when one is studying heterogeneity in immune cell responses and rare cell behavior is expected. In our approach, by collecting droplets off-chip we maintain the potential of unlimited throughput and through the encapsulation in microgels, we maintain spatial information while allowing fluorescent staining of pairs for a high number of parameters. To further clarify these advantages over other applications in the field we have revised the introduction.
Changes in the revised version:
P1-2: “In line with that, several sub-populations of CTLs have been identified based on secreted cytokines and the presence of polarizing cytokines4,5, suggesting that specific subsets of CTLs are relevant for immunity. Since interrogation of each individual cell is needed to truly identify CTLs of interest, the field has recently moved from conventional bulk experiments to single-cell approaches. Over the past two decades, single-cell technologies have emerged with the goal to dissect cellular heterogeneity and interrogate relevant sub-populations6–10. To efficiently and reproducibly probe the heterogeneity upon CTL activation, microfluidic tools were developed to allow precise manipulation and compartmentalization of single cells in small volumes (pico- to nano-liter)11. Thereby providing highly controlled environments acting like bioreactors to efficiently activate CTLs12,13. Although these efforts to pair CTLs either with target cells or APCs in microwell- and microtrap-based devices have proven to be effective, they are often restricted by a limited throughput since pairs are captured on-chip and sample size is thus restricted by the dimensions of the device12,14. Droplet-based microfluidics has the major advantage that the sample size can be increased by longer running times thereby yielding higher throughputs15–18, which is especially important when one is studying heterogeneity in immune cell responses and rare cell behavior is expected. Previous work aimed at pairing and studying CTL responses in droplets yielded interesting findings at single-cell resolution19,20. However, these studies were often limited to the measurement of only a few parameters through microscopy imaging. As various factors, including both membrane marker expression and cytokine secretion, have been previously used to score CTL quality, combining these will allow for true interrogation of the heterogenous CTL response1–3. To overcome both the challenge of limited throughput as well as allowing measurement in a multiparameter fashion, hydrogels can be incorporated during droplet production to obtain “microgels”, in which cells can be paired. These will maintain spatial coupling of single-cell pairs while allowing subsequent cell profiling using flow cytometry, facilitating a multiparameter measurement21. Hence, microgels could pave the way for novel and robust analytic tools to study single CTL and APC interactions12,22,23.”
Comment 3: The cell encapsulation proportion in the microdroplet looks low in Figure 1b and Figure 2a. How to improve the encapsulation proportion of single cells, just by increasing the cell density does not seem to guarantee that each droplet contains a single cell? How to separate the microdroplets that are encapsulated with cells from unencapsulated cell droplets during testing? The number of microdroplets enveloping cells shown in Figure 2a is quite different from the statistical value in Figure 2b. Please provide more accurate optical photos.
Response to comment 3: The process of encapsulating cells inside droplets is random and follows a Poisson distribution. As a result, the mean number of cells per droplet, λ, is determined by the number of cells in the initial solution and the volume of an individual droplet. Droplet occupancy can therefore be tuned by changing the initial cell density or the droplet size (via the flow rate values). Although a higher λ might be advantageous to increase the number of low-frequency events, it will also result in a higher frequency of droplets containing multiple cells. A lower λ is therefore preferable in this experimental setup as it will yield more single-cell encapsulation, but additionally also yielding more empty droplets. Then again, empty droplets are not an issue for performing downstream analysis. In our case with studying cell behavior in aqueous droplets these empty droplets will disappear in the de-emulsified solution without affecting cells, and in the case of microgels they can easily be filtered out during analysis according to their specific forward scatter (FSC-A) signature as demonstrated in Supplementary figure 2.
We agree with the reviewer that Figure 2a does not represent the statistical distribution shown in Figure 2b. This image was selected for aesthetic reasons and visualization of Dynabeads paired with T-cells. We have therefore included an additional representative image of a larger number of droplets in the supplementary figures.
Changes in revised version:
Supplementary figure 2: was added
P6: “By switching from soluble stimuli to aAPCs, the encapsulation efficiency is altered but the prevalence of all combinations of encapsulation still followed the predicted Poisson distribution (Figure 2B and Supp. Figure 2).”
Comment 4: An optical photo of staining should be shown in Figure 3 to prove that the pore size of agarose hydrogels allowed diffusion of fluorophore-conjugated antibodies to ensure that CTL activation in microgels.
Response to comment 4: We agree with this suggestion from the reviewer. Obviously, analysis with flow cytometry can already provide proof of this concept, e.g. staining of CTLs in droplets with antibodies recognizing an expressed protein (positive control) and staining with antibodies for a protein that is not expressed (negative control). When CTLs in microgels display a fluorescent signal for the protein that is expressed and not for the negative control antibodies, this is a direct implication that antibodies can diffuse in and out microgels for staining purposes. Nevertheless, we indeed agree that it would be a nice addition to have visual confirmation that illustrate fluorescent antibody staining of encapsulated cells is possible and similar to staining of cells outside of the microgel. Therefore, we performed an additional set of experiments described above and have added a supplementary figure with fluorescent images of cells in microgels and outside of microgels stained with the largest used fluorophore (PE), and the corresponding flow cytometric readout. Additionally, we referred to this in the text.
Changes in revised version:
P7: “Additionally, microscopic images showed that staining was cell-specific and did not differ between encapsulated and un-encapsulated cells (Supp. Figure 4). Even for IL-2 detection, which uses the largest fluorophore-antibody conjugate42, only a small difference could be observed in fluorescent intensity. Thus, microgel encapsulated cells are compatible with fluorescent staining for flow cytometric analysis. Future additions of fluorophores can be reliably incorporated to extend the antibody panel for multidimensional analysis
Supplementary Figure 4: Was added
Comment 5: In page 2, line 88-89, the authors described that “After bonding, the channels were treated with 5% perfluorooctyltriethoxysilane in HFE-7500 (both from Fluorochem)”. What does the HFE-7500 mean? Why does the channel need to be treated with 5% perfluorooctyltriethoxysilane in HFE-7500?
Response to comment 5: The reviewer is correct that some additional clarification is needed. HFE-7500 is the fluorinated oil that can be considered the golden-standard for the continuous phase in droplet microfluidics as it is biocompatible and allows exchange of O2 and CO2. Additionally, the channels of our device need to be hydrophobic to prevent the aqueous phase from interacting with the channel walls affecting droplet formation. This is also common practice for droplet microfluidics, but to clarify we have included this in the material and methods.
Changes in revised version:
P2: “After bonding, the channels were treated with 5% perfluorooctyltriethoxysilane in HFE-7500 fluorinated oil (both from Fluorochem) in order to make channel walls hydrofobic, incubated for 1 hour at 65°C, flushed again with HFE-7500, and incubated overnight at 65°C for thermal bonding.”
Comment 6: In page 3, line 112, the authors demonstrated that “Cells were injected at a concentration of 4*106 cells/ml”. “…4*106 cells/ml…” should be “… 4*106 cells/ml …”. Please pay attention to check the details and the specification of writing.
Response to comment 6: We thank the reviewer for pointing this out, we have made the necessary corrections in our new and improved manuscript.
Changes in revised version:
P3: “Cells were injected at a concentration of 4*106 cells/ml. When Dynabeads were included, these were injected at a concentration of 1*107 particles/ml.”

Reviewer 2 Report
Reviewer comments for the article no. micromachines-1976027, entitled “A microfluidic approach for probing heterogeneity in cytotoxic T-cells by cell pairing in hydrogel droplets”.
General evaluation:
In this study, the Authors fabricated a microfluidic device optimized to study the single cell interactions between APCs and CTL by using the physical thermoregulatory properties of agarose-based microgels.
This article is interesting however there are several compulsory major points, described below, that need to be fully clarified also by the addition of new experiments. In addition, the english grammar should be accurately revisited. Finally, the depicted conclusions are not supported by the results and for this reason the discussion must be fully revisited.
Major points:
1) The Authors provided poor rationales and no information about the reason why they employed the aAPCs rather than normal APCs from human healthy donors as a source of natural (rather than artificial) APCs to study the CTL-APCs interactions. This reviewer strongly recommends to extensively clarify these issues and, in parallel to aAPCs to present results made with experiments done by using “normal” APCs.
2) Alongside the experimental design proposed in the point 1, the Authors must confirm these experiments with a mouse model system. For example, the can use CTLs and APCs from OT-I and OT-II C57BL/6 mice (spleen, thymus, lymph nodes). T cell from OT-I (CD4+T cells) mice will be engage in an active antigen presentation when stimulated by an OVA peptide. Similarly, in OT-II mice the OVA-restricted presentation will be made by CD8+T cells. This provide a strong advantage to distinguish the behaviour of CTL-APC interactions in each organ, in an OVA-restricted fashion of APC presentation.
3) Figure 1A represents a schematization of the device used in this study. However, this layout is very poor in terms of details such as measurements, chambers, etc. Please re-design it by taking into account these detail, possibly making a 3D rendered layout structure. In parallel, an exact and detailed geometry of the device (and their inner substructures, if any) must be shown as layout.
4) To me, it is unclear the exact way, in Figure 2, CTLs (herein CD8+T cells) & aAPCs are added into the chip. Please draw a layout showing this, and add a brief supplementary video to easily display this CTLs and aAPCs entering into the device.
5) An important question to deepen in Figure 2A is to identify APC/CTL doublets that are effectively in contact. For example, the APC in the droplet at the top of panel A seems to be away (and then not in contact) from the CTL. Author should then include, as additional conditions in the panel B the doublets where APC is really in contact with CTL to distinguish by those where APC is NOT in contact with the CTL. The experiments should be performed on the droplets where APCs have effectively contacted CTLs.
6) The comment 5 must also be applied to experiments shown in Figure 4, in order to distinguish droplets with “true” APC/CTL contact from those with “false” contact. Moreover, in Figure 4B it is not explained the way Authors do physically distinguish “Contact” (Cyan color) and “No Contact” (yellow color) doublets. This reviewer strongly suggests to add a supplementary time lapse video to demonstrate that in APC/CTL doublets confined in droplets are able to effectively made a cell-cell contact.
Author Response
Reviewer #2:
General comment: In this study, the Authors fabricated a microfluidic device optimized to study the single cell interactions between APCs and CTL by using the physical thermoregulatory properties of agarose-based microgels.
This article is interesting however there are several compulsory major points, described below, that need to be fully clarified also by the addition of new experiments. In addition, the english grammar should be accurately revisited. Finally, the depicted conclusions are not supported by the results and for this reason the discussion must be fully revisited.
Response to general comment: We would like to thank the reviewer for her/his interest in this manuscript as well as its relevant comments which should help improve the manuscript’s quality. The reviewer raised several aspects to add or revise and we will address each of them in the following:
Comment 1: The Authors provided poor rationales and no information about the reason why they employed the aAPCs rather than normal APCs from human healthy donors as a source of natural (rather than artificial) APCs to study the CTL-APCs interactions. This reviewer strongly recommends to extensively clarify these issues and, in parallel to aAPCs to present results made with experiments done by using “normal” APCs.
Response to comment 1: We agree with the reviewer that looking at the heterogenous interaction between human donor blood derived CTLs and APCs would be highly interesting and relevant. However, in this manuscript we set out to focus on CTL heterogeneity specifically by developing a platform that allows us to stimulate individual cells with high-throughput and reveal heterogeneity under controlled conditions. We decided to benchmark our platform and probe heterogeneity of CTLs by using artificial APCs (aAPCs) to activate CTLs. aAPCs come in various forms, shapes and sizes, but most researchers use the commercially available Dynabeads to study CTL activation by aAPCs (10.1186/1479-5876-8-104, 10.1016/j.celrep.2020.107574 and 10.1089/hgtb.2014.051). Dynabeads are well characterized, and homogenous micron-sized beads decorated with anti-CD3 and anti-CD28 antibodies which provide stimulatory cues for CTLs, i.e. T cell receptor complex activation by anti-CD3 (signal 1) and costimulatory molecule activation by anti-CD28 (signal 2). One of the major advantages of using aAPCs (Dynabeads here) is that there is no heterogeneity present in this system, in contrast to natural APCs that are highly heterogenous by nature and origin, or display functional heterogeneity that is induced upon APC activation. Especially as we aimed to compare the efficiency of activation between soluble chemical stimulation and contact-mediated activation, the influence of APC heterogeneity had to be excluded. Otherwise, we potentially would mix an unknown APC heterogeneity and the effect that might have on inducing an unknown CTL heterogeneity. By no means this implies that we are not interested in pursuing the effect of natural APC on CTL activity, but we believe that this is beyond the scope of the current manuscript as this will be largely dependent on the natural APC type, stimulus used for APC stimulation, and mechanism for antigen loading on APC. Having said that we did perform an additional experiment where we demonstrate the potency of our platform for the investigation of natural cell-cell pairs and have therefore added flow cytometric data of pairing of CTLs and monocytes to illustrate how the platform can be adjusted to perform experiments the reviewer refers to.
Changes to revised version:
P9: ”Besides pairing of CTLs with beads, the platform can easily be adjusted to pair CTLs with another cell type. This would merely require the incorporation of an additional membrane marker staining, cell pairs can then be selected by gating for double positive events (Supp. Figure 6)”
Supplementary figure 6: Was added
Comment 2: Alongside the experimental design proposed in the point 1, the Authors must confirm these experiments with a mouse model system. For example, the can use CTLs and APCs from OT-I and OT-II C57BL/6 mice (spleen, thymus, lymph nodes). T cell from OT-I (CD4+T cells) mice will be engage in an active antigen presentation when stimulated by an OVA peptide. Similarly, in OT-II mice the OVA-restricted presentation will be made by CD8+T cells. This provide a strong advantage to distinguish the behaviour of CTL-APC interactions in each organ, in an OVA-restricted fashion of APC presentation.
Response to comment 2: From an immunological point of view, it would be interesting and informative to understand whether APCs and CTLs derived from various lymphoid organs display distinct behavior and induction of OVA specific immune responses. However, in order to successfully perform these types of experiments it is crucial that one has access to animal facilities and the suggested OT-I and OT-II mice. Moreover, in the Netherlands the use of animal experiments is tightly regulated and requires applications of research proposals. To be able to carry out an animal experiment, the researcher first needs to apply for a project license. The application must include a detailed description of the research project, in which the researcher makes it clear why an animal experiment must be used and not an alternative approach. The applicant is legally obliged to describe why an approach that does not use animal testing is not possible, and why it is not possible to carry out the experiment with fewer animals or with less suffering for the animals. Only researchers who have the relevant training (an ‘Article 9 status’) and are employed by an institution that is licensed to carry out animal experiments (a licensee) may apply to conduct an animal experiment. However, even if a researcher considers an animal experiment to be necessary, there is a long way to go before the experiment can start. So, although we agree that the suggested experimental design would be interesting, the use of mouse models is far beyond the scope of our research.
Comment 3: Figure 1A represents a schematization of the device used in this study. However, this layout is very poor in terms of details such as measurements, chambers, etc. Please re-design it by taking into account these detail, possibly making a 3D rendered layout structure. In parallel, an exact and detailed geometry of the device (and their inner substructures, if any) must be shown as layout.
Response to comment 3: We thank the reviewer for their remark and agree that details of dimensions are missing. As the layout was already a direct copy of our design of the device we included all relevant dimensions. As the height of the devices channels and chambers is equal everywhere we did not include a 3D model, but we did clarify this in the figure’s description.
Changes in revised version:
Figure 1: Panel A has been altered.
Figure 1 description: “Layout of the microfluidic device used for droplet production including popped-out schematic of droplet formation, scale bar represents 500 µm. Layout is a direct copy of the photolithography mask design, and channel height is equal throughout.”
Comment 4: To me, it is unclear the exact way, in Figure 2, CTLs (herein CD8+T cells) & aAPCs are added into the chip. Please draw a layout showing this, and add a brief supplementary video to easily display this CTLs and aAPCs entering into the device.
Response to comment 4: We thank the reviewer for pointing out this unclarity. To answer this comment, we have performed a couple of experiments and by using a high-speed camera we were able to generated videos during droplet production at the production junction in the device. We have adjusted Figure 2 accordingly by adding an illustrative layout of droplet formation and have included a supplementary video of CTLs and aAPCs entering the droplets during droplet formation. We would kindly like to refer the reviewer to our previously published sample loading method which was cited in the materials and method of the current manuscript. Briefly, this approach utilizes pipette tips attached to a microfluidic pump system. The pipette tip compartment allows the aspiration of solutions containing cells, beads or stimuli. And after insertion of the pipette tip onto the microfluidic device hydraulic pressure is used to inject the solution into the channels (10.3791/57848-v).
Changes in revised version:
Figure 2: Panel A has been added.
Figure 2 description: “Schematic of droplet formation when CTLs are paired with Dynabeads.”
Supplementary material: A video of the droplet formation point has been added.
P6: “To mimic the first two signals of this cell-cell contact-dependent activation in our platform we aimed to co-encapsulate CTLs with aAPCS (Figure 2A and B and Supp. Video 1), which contain anti-CD3 and anti-CD28 antibodies and are routinely used to activate T-cells both in bulk as well as at single-cell level38,39.”
Comment 5: An important question to deepen in Figure 2A is to identify APC/CTL doublets that are effectively in contact. For example, the APC in the droplet at the top of panel A seems to be away (and then not in contact) from the CTL. Author should then include as additional conditions in the panel B the doublets where APC is really in contact with CTL to distinguish by those where APC is NOT in contact with the CTL. The experiments should be performed on the droplets where APCs have effectively contacted CTLs.
Response to comment 5: We thank the reviewer for this sharp observation. The image shown in Figure 2A (as well as those in Figure 1B, the first panel of Figure 3C and first panel of Figure 4A) is taken within 10 seconds after droplet production by sampling some fresh produced droplets on a glass slide and viewing them under a microscope. This means that cells are still floating around freely and have not gotten the chance to interact yet. We capture the images of droplets in this way as the fluorinated oil in the continuous phase is highly volatile and there is thus only a short timeframe in which images can be made. We recognize the concerns raised here by the reviewer and therefore want to illustrate how after droplet production within a very short timeframe cells will interact within aqueous droplets. Therefore, we performed additional experiments where we produced aqueous droplet with CTL/aAPC pairs and collected them in observation chambers (to avoid evaporation of oil) and monitored them over time. By the time these droplets were placed under the microscope however, all cells had already latched onto the aAPCs. We have included a supplementary video showing this to illustrate that in aqueous droplets interaction of cell pairs is not a problem (as frequently observed in literature; 10.1021/acsomega.0c03264, 10.1038/s41598-021-96609-9).
Changes in revised version:
Supplementary material: A video of the first 30 minutes after collecting aqueous droplets with CTL and aAPC pairs has been added.
P6: “After in-droplet co-culture of single cell pairs and verification of CTL/aAPC interaction based on cell morphology (Supp. Video 2), we assessed CTL activation and observed heterogeneous expression of CD69 and CD25 (Figure 2C), and very little cytokine secretion in only a small percentage of CTLs (Figure 2D).”
Comment 6: The comment 5 must also be applied to experiments shown in Figure 4, in order to distinguish droplets with “true” APC/CTL contact from those with “false” contact. Moreover, in Figure 4B it is not explained the way Authors do physically distinguish “Contact” (Cyan color) and “No Contact” (yellow color) doublets. This reviewer strongly suggests adding a supplementary time lapse video to demonstrate that in APC/CTL doublets confined in droplets are able to effectively made a cell-cell contact.
Response to comment 6: In contrary to cell-pairing in aqueous droplets we were more concerned initially about cell interaction after encapsulation in the slightly more viscous agarose solutions. We therefore aimed to quantify the number of cells that were interacting after overnight incubation and the ones that were not, as depicted in Figure 4A & 4B. We distinguished the “Contact” and “No contact” visually. We added a supplementary timelapse video similar to the one mentioned in the response to comment 5, but this time with agarose solution droplets. The video nicely illustrates that cell contact is made within a few minutes, warranting CTL activation.
Changes in revised version:
Supplementary material: A video of the first 30 minutes after collecting agarose droplets with CTL and aAPC pairs has been added.
P9: “Additional temporal monitoring over the first 30 minutes after encapsulation was performed as well and displayed cell latching onto aAPCs within minutes, demonstrating that cell-cell interaction takes place, even in agarose droplets (Supp. Video 3).”

Reviewer 3 Report
questions:
1, Will the oil phase affect the viability of the cells?
2, The scale bar should be labelled in Figure 4A
Author Response
Reviewer #3:
General comment: None
Comment 1: Will the oil phase affect the viability of the cells?
Response to comment 1: We thank the reviewer for the reading of our manuscript. The fluorinated oil HFE-7500 as used in our experiments is the golden standard for cell cultures in droplets and has been designed specifically for it by allowing O2 and CO2 exchange. Its effect on cell viability has been tested (10.1007/s10544-016-0137-0) and is regarded as negligible. We do underscore the importance for knowing cell viability in the experimental system and to eliminate any unlikely effect of cell viability on our data we always use viability staining which allows exclusion of non-viable cells from the data analysis.
Comment 2: The scale bar should be labelled in Figure 4A
Response to comment 2: The reviewer is right, therefore we have adjusted accordingly.

Round 2
Reviewer 1 Report
The revised manuscript has been sufficiently improved to warrant publication in Micromachines.
Reviewer 2 Report
The Author significantly improved the manuscript and clearly answered to all the questions posed by this Reviewer. This manuscript is now acceptable for publication in Cells.